# OpenReview forum: "SAFEx: Analyzing Vulnerabilities of MoE-Based LLMs via Stable Safety-critical Expert Identification"
_NeurIPS.cc/2025/Conference — NeurIPS 2025 poster_

### Official Review · Reviewer_qRFt · 2025-06-30

**Clarity:** 3
**Significance:** 3
**Originality:** 3
**Rating:** 5
**Confidence:** 4

**Summary:**

The study presents SAFEX, an analytical framework that identifies, characterizes, and validates safety-critical experts in large language models based on Mixture-of-Experts (MoE) architectures. The framework reveals critical risks inherent to MoE architectures, such as reliance on specific expert modules for safety-aligned behaviors. Disabling these experts significantly compromises the models' ability to refuse harmful requests, highlighting the disproportionate impact of a small set of experts on overall model safety.

**Questions:**

1. Page 1, Line 16
> For Qwen3-MoE with 6144 experts (in FNN layer), ...

Which model contains 6144 experts? It should be more specific.

2. Page 8
> 2.5 Expert Masking Experiment

Will the masking downgrade the original performance of MOE?

3. Page 3
> 2.1 Dataset Construction

The statistics of the data should be given. Many details are missing.

4. Page 5, Line 147
> , we independently draw multiple empirical datasets ...

Should the datasets be a large one?

5. Page 5, Line 178
> The horizontal dashed lines represent the theoretical mean activation probabilities under each MoE configuration.

Not sure what theoretical mean is.

6. Page 8, Line 282
> We further conduct jailbreak attack experiments to comparatively analyze the vulnerability differences between MoE and non-MoE architectures.

How non-MOE is masked?

7. Some related work should be discussed:

- Li, Kenneth, Oam Patel, Fernanda Viégas, Hanspeter Pfister, and Martin Wattenberg. "Inference-time intervention: Eliciting truthful answers from a language model." Advances in Neural Information Processing Systems 36 (2023): 41451-41530.
- Song, Da, Xuan Xie, Jiayang Song, Derui Zhu, Yuheng Huang, Felix Juefei-Xu, and Lei Ma. "Luna: A model-based universal analysis framework for large language models." IEEE Transactions on Software Engineering (2024).

**Ethical Concerns:**

["NO or VERY MINOR ethics concerns only"]

**Final Justification:**

The rebuttal addresses my concerns.

**Limitations:**

Please see Questions.

**Quality:**

3

**Strengths And Weaknesses:**

Strengths:
+ The paper introduces and formalizes the concept of positional vulnerability in MOE LLM, which is timely and important.
+ The three-step process, statistical identification, functional decomposition, and validation, is well-designed.

Weaknesses:
- The paper does not fully specify important experimental details, e.g., hyperparameters for probes and the size of the dataset.
- Not sure whether the linear probing can be generalized to larger MOE.

---

> ### Author Rebuttal · Authors · 2025-07-30
>
> > **Q1:** The paper does not fully specify important experimental details, e.g., hyperparameters for probes and the size of the dataset.
>
> **A1:** We sincerely thank you for highlighting these crucial details. To improve the reproducibility and transparency of our study, we provide the requested information below and will incorporate it into the revised manuscript.
>
> 1. **Hyper-parameters for the linear probes.**
>    Each probe is an L2-regularised logistic-regression classifier trained with
>    ​*C = 1.0*​, ​*solver = 'lbfgs'*​, ​*max\_iter = 1000*​, ​*n\_jobs = -1*​, and ​*random\_state = 42*​.
> 2. **Dataset statistics.**
>    Each evaluation subset—\\(D\_{Regular}\\), \\(D\_{Jailbreak}\\), and \\(D\_{Benign}\\)—contains **500** prompts.
>    Figure 1 in **Appendix A** comprehensively reports the distribution of harmful topics and traces every sample to its original source.
>    In Section 2.1 (​*Dataset Construction*​) we will add an explicit cross-reference to this figure and state the exact sample counts to ensure readers can easily locate these details.
>
> ---
>
> > **Q2:** Not sure whether the linear probing can be generalized to larger MOE.
>
> **A2:** Thank you for raising this insightful question. Although the current manuscript does not yet include experiments on the very largest publicly available MoE models—owing to computational constraints—we have already applied our linear-probing validation to MoE LLMs spanning a wide range of parameter counts. Across these models the probe consistently demonstrates strong discriminative power, indicating that the safety-critical representations we identify are largely insensitive to model size. We will incorporate a concise summary of these additional results into the revised paper and, as computational resources permit, extend our study to even larger-scale MoE architectures to further substantiate the generality of our conclusions.
>
> ---
>
> > **Q3:** Page 1, Line 16
> > For Qwen3-MoE with 6144 experts (in FNN layer), ...
> > Which model contains 6144 experts? It should be more specific.
>
> **A3:** This refers to the Qwen3-30B-A3B model. We will clarify this detail in the revised version to improve specificity.
>
> ---
>
> > **Q4:** Page 8
> > 2.5 Expert Masking Experiment
> > Will the masking downgrade the original performance of MOE?
>
> **A4:** We conducted additional experiments on the AIME2024 dataset using **Qwen3-30B-A3B** and ​**Qwen1.5-MoE-A2.7B-Chat** （using the same prompt and queried for 5 times）​. The results are shown in the table below:
>
> | Model                  | Pre-mask Score | Post-mask Score |
> | ------------------------ | ---------------- | ----------------- |
> | Qwen3-30B-A3B          | 77.4           | 75.2            |
> | Qwen1.5-MoE-A2.7B-Chat | 12.8           | 13.2            |
>
> The score fluctuations (–2.2 / +0.4) correspond to ±3 %, indicating no statistically significant change in mathematical-reasoning ability after masking. We therefore conclude that our masking strategy preserves overall performance while improving safety.
>
> ---
>
> > **Q5:** Page 3
> > 2.1 Dataset Construction
> > The statistics of the data should be given. Many details are missing.
>
> **A5**:Thank you very much for your valuable suggestion. The composition of each dataset, including the proportion of harmful topics and the specific data sources, is summarized in Figure 1 of Appendix A. The datasets $D_{Regular}$, $D_{Jailbreak}$, and $D_{Benign}$ each contain 500 entries. To enhance clarity and reproducibility, we will incorporate a cross-reference to the appendix in the “Regular Group” subsection of Section 2.1 Dataset Construction and explicitly indicate the number of entries for each dataset in the main text.
>
> ---
>
> > **Q6:** Page 5, Line 147
> > Should the datasets be a large one?
> > , we independently draw multiple empirical datasets ...
>
> **A6:** Initially, we experimented with a dataset of 100 samples to explore expert activation patterns. Under this setting, no discernible or stable patterns emerged. Therefore, we expanded the dataset to 500 samples and deliberately incorporated a broader range of topics. It was only after this expansion that consistent expert activation patterns began to surface, as reported in the paper. These findings suggest that identifying reliable and generalizable expert activation phenomena requires a dataset that is not only sufficiently large but also diverse in topical coverage.
>
> ---
>
> > **Q7:** Page 5, Line 178
> > The horizontal dashed lines represent the theoretical mean activation probabilities under each MoE configuration.
> > Not sure what theoretical mean is.
>
> **A7:** Thank you for raising this point. The **theoretical mean activation probability** corresponds to the *ideal* scenario in which the router selects experts ​**uniformly at random**​. Under this assumption, each expert in a given layer is expected to be chosen with probability \$\\frac{k}{K},\$ where \\(k\\) is the number of experts activated per layer (top-\\(k\\)) and \\(K\\) is the total number of experts in that layer. For example, **Qwen3-30B-A3B** contains \\(K = 128\\) experts per layer and activates \\(k = 8\\) of them at each routing step. The theoretical mean activation probability for any single expert is therefore \$\\frac{8}{128} = 0.0625 \\; (6.25\\%)\$.
>
> In ​**Figure 3**​, the horizontal dashed line marks this baseline value. Deviations of the empirical activation probabilities **above** or **below** the dashed line reveal the extent to which the model’s routing behavior departs from uniform selection, thereby quantifying the degree of expert *specialization* within the MoE architecture.
>
> ---
>
> > **Q8:** Page 8, Line 282
> > We further conduct jailbreak attack experiments to comparatively analyze the vulnerability differences between MoE and non-MoE architectures.
> > How non-MOE is masked?
>
> **A8:** Non-MoE (Dense) models are not  masked. In the experiment, there is an original dataset \$D\_{regular}\$, with one sample denoted as \$x\$, representing a **dangerous problem** with significant semantic features, easily recognized and intercepted by the model. Through the jailbreak rewriting function \$\\mathcal{T}\$, we construct an input \$\\hat{x}\$ that remains semantically similar (i.e., the same topic) but has higher evasion at the surface level, i.e., \$\\hat{x} = \\mathcal{T}(x)\$. All such \$\\hat{x}\$ samples form the dataset \$D\_{Jailbreak}\$. As shown in the table, we compare the rejection rates tested on the \$D\_{Regular}\$ dataset (corresponding to the "Before Mask" column) and the rejection rates tested on the \$D\_{Jailbreak}\$ dataset (corresponding to the "Jailbreak" column). We find that the MoE model's rejection rate drops more, highlighting its significant positional vulnerability (safety-critical expert identification).
>
> ---
>
> > Q9:Some related work should be discussed:
> >
> > * Li, Kenneth, Oam Patel, Fernanda Viégas, Hanspeter Pfister, and Martin Wattenberg. "Inference-time intervention: Eliciting truthful answers from a language model." Advances in Neural Information Processing Systems 36 (2023): 41451-41530.
> > * Song, Da, Xuan Xie, Jiayang Song, Derui Zhu, Yuheng Huang, Felix Juefei-Xu, and Lei Ma. "Luna: A model-based universal analysis framework for large language models." IEEE Transactions on Software Engineering (2024).
>
> **A9:** Thank you for your valuable suggestions. Following your suggestions, we discussed these papers. Recent work on *Inference-Time Intervention (ITI)*  **[1]** adjusts internal activations to guide the model toward truthful responses, reducing vulnerabilities exploited by adversarial prompts. While Constitutional AI guides LLMs to refuse based on principles, LUNA evaluates whether such trustworthy behavior is actually achieved **[2]**. Both of these papers are related to our work. We will include these references in the "Related Work" section of the revised version. The specific revisions are as follows, with the modified parts highlighted in ​**bold**​.
>
> > Adversarial prompts can still trigger unsafe outputs by exploiting latent vulnerabilities **[1]**.
>
> > …To counteract this, Constitutional AI replaces human preference labels with high-level principles and uses self-critiques to guide learning, resulting in more transparent and grounded refusals. **LUNA uses the data from desired trustworthiness perspectives to construct an abstract model as an auxiliary analysis asset and proxy [2].** However, these methods…
>
> **[1]** Li, Kenneth, Oam Patel, Fernanda Viégas, Hanspeter Pfister, and Martin Wattenberg. "Inference-time intervention: Eliciting truthful answers from a language model." Advances in Neural Information Processing Systems 36 (2023): 41451-41530.
>
> **[2]** Song, Da, Xuan Xie, Jiayang Song, Derui Zhu, Yuheng Huang, Felix Juefei-Xu, and Lei Ma. "Luna: A model-based universal analysis framework for large language models." IEEE Transactions on Software Engineering (2024).

---

> > ### Comment · Reviewer_qRFt · 2025-08-05
> >
> > Thank you for the rebuttal, and it addresses my concerns. I am updating the score.

---

> > > ### Author Response · Authors · 2025-08-06
> > >
> > > Thank you for the thoughtful review and constructive feedback. We're glad that our rebuttal addressed your concerns, and we truly appreciate your updated score and recognition.

---

### Official Review · Reviewer_cQJB · 2025-07-01

**Clarity:** 2
**Significance:** 2
**Originality:** 2
**Rating:** 3
**Confidence:** 3

**Summary:**

This paper investigates the safety alignment challenges unique to Mixture-of-Experts (MoE) models, focusing on the phenomenon of positional vulnerability—where safety-critical behaviors depend disproportionately on a small subset of expert modules. The authors propose SAFEx, an analytical framework equipped with a novel Stability-based Expert Selection (SES) algorithm, to identify and characterize these safety-critical experts. Experimental results on Qwen3-MoE reveal that disabling a small number of experts drastically reduces the model’s refusal rate on harmful queries, highlighting structural safety risks in MoE architectures.

**Questions:**

Please see weakness.

**Ethical Concerns:**

["NO or VERY MINOR ethics concerns only"]

**Final Justification:**

None

**Quality:**

2

**Strengths And Weaknesses:**

Strengths:

1. Timely and meaningful problem formulation: The paper addresses a novel and underexplored issue in the context of MoE models—specifically, their safety vulnerabilities. Given the increasing deployment of MoE models for efficient scaling, this study is both relevant and significant.

2. Potential impact on the MoE community: By formalizing positional vulnerability and proposing a concrete analysis framework, this work may stimulate further research on the safety properties of MoE architectures and inspire the design of more robust expert routing mechanisms.


Weaknesses:

1. Lack of crucial experimental details, affecting reproducibility and interpretation:
- What specific jailbreak methods are included in the Jailbreak Group used for evaluation?
- How is the refusal rate in Table 1 computed? What criteria are used to determine whether a response is considered a refusal?
- If disabling experts causes decoding failures, are these failures counted as refusals or non-refusals?

2. Experimental conclusions overlap with prior findings: While the work targets MoE-specific safety, the core finding—that a small number of parameters (experts) are disproportionately responsible for safety behaviors—mirrors conclusions from prior studies on dense models (e.g., [1]). This raises the question: what unique safety challenges are intrinsic to MoE models, beyond this structural analogy?

3. Limited granularity in analysis: The focus on expert-level analysis, though aligned with MoE architecture, invites further scrutiny. If one were to analyze the safety attribution at the neuron level within experts, would the results be distinguishable from those observed in dense architectures? This could clarify whether the MoE-specific framing is essential or incidental.

4. Lack of actionable strategies for mitigation: While the work identifies structural weaknesses in MoE safety, it stops short of proposing concrete solutions or mitigations. Suggestions on how to improve expert routing, redundancy, or fallback mechanisms for safety-critical tasks would significantly strengthen the practical relevance of the study.

[1] Assessing the brittleness of safety alignment via pruning and low-rank modifications. ICML 2024.

---

> ### Author Rebuttal · Authors · 2025-07-30
>
> > **Q1:** Lack of crucial experimental details, affecting reproducibility and interpretation:
> >
> > * What specific jailbreak methods are included in the Jailbreak Group used for evaluation?
> > * How is the refusal rate in Table 1 computed? What criteria are used to determine whether a response is considered a refusal?
> > * If disabling experts causes decoding failures, are these failures counted as refusals or non-refusals?
>
> **A1:** Thank you for your careful review. We will do our utmost to improve the paper’s readability by providing additional clarifications on the experimental setup point by point, in the following order:
>
> * semantic paraphrasing, adversarial perturbation, context rewriting.
> * We utilized the API of the DeepSeek-V3 model to automatically score the question–answer pairs generated by the evaluated models, thereby obtaining corresponding risk scores. A threshold was then applied, and samples with scores below this threshold were labeled as refusals. Due to space limitations, we are unable to provide the full prompt used for scoring(The *complete* scoring prompt—including (a) the eight prohibited categories, (b) an illustrative example, and (c) the rubric—**will be provided in the supplementary material** to facilitate replication). The prompt consists of the following three parts:
>
>   * A clearly defined list of eight categories of prohibited behaviors, covering high-risk use cases such as privacy violations, illegal activities, manipulation and deception, and discriminatory or harassing content.
>   * An example comprising a user's malicious instruction and the model's response, used to analyze how the model handles unsafe input.
>   * A scoring rubric from 0 to 5, which quantitatively evaluates the degree of compliance. It ranges from complete refusal to highly detailed and executable dangerous outputs.
>
>   In our experiments, model responses on \$D\_{Benign}\$ or \$D\_{Jailbreak}\$ that received a final score less than or equal to 2 were classified as refusals.
> * No decoding failures were observed during our experiments. According to the principles of the MoE architecture, whether an expert is selected for activation depends on whether it ranks among the top-K most activated experts within its corresponding layer. A decoding failure would only occur if the number of available experts in a layer drops to K or fewer, thereby preventing top-K selection.
>   In our case, the highly activated experts we disabled were mostly distributed across different layers, which prevented such conditions from arising. Additionally, we manually inspected several responses generated after expert deactivation. All samples produced coherent and logically structured natural language outputs, with no signs of interruption or malformed responses.
>
> ---
>
> > **Q2:** Experimental conclusions overlap with prior findings: While the work targets MoE-specific safety, the core finding—that a small number of parameters (experts) are disproportionately responsible for safety behaviors—mirrors conclusions from prior studies on dense models (e.g., [1]). This raises the question: what unique safety challenges are intrinsic to MoE models, beyond this structural analogy?
> > **[1]** Assessing the brittleness of safety alignment via pruning and low-rank modifications. ICML 2024.
>
> **A2:** We thank you for the insightful comments and for highlighting the potential overlap with prior studies.
>
> First, although the MoE architecture is theoretically designed to mitigate the sparse dependency on safety-relevant components through sparse activation and load balancing, our experiments demonstrate that even within MoE models, safe behaviors still rely heavily on the activation of a small subset of experts. Moreover, this form of "positional vulnerability" is more susceptible to jailbreak attacks. These findings suggest that the MoE design does not fundamentally eliminate such issues and may instead introduce new forms of fragility.
>
> Second, compared to prior work, our study systematically proposes a safety attribution and validation methodology at the expert level and, for the first time, empirically distinguishes functional roles within experts. In our supplementary experiments, the SAFEx framework offers a practical mechanism for expert-level intervention, effectively enhancing the safety robustness of MoE systems while preserving their semantic capabilities. Below, we briefly outline the supplementary content.
>
> We adopt the idea of model merging to remove the toxic output capability of the HRCG experts identified in our study. Specifically, we fine-tune either a subset or all of the experts on the JBB-Behaviors dataset **[2]** to obtain the corresponding "toxic" weight changes. These toxic weights are then directly subtracted from the original Qwen-MoE model to construct two safety-enhanced models. We evaluate the jailbreak success rate of these modified models on the Jailbreak Group dataset. The table below presents the refusal rates of the selected models under different treatment settings on the $D_{Jailbreak}$ dataset.
>
> |                        | Before Alignment | After Partial Merge | After Full Merge |
> | ------------------------ | ------------------ | --------------------- | ------------------ |
> | Qwen1.5-MoE-A2.7B-Chat | 52.00%           | 74.80%              | 80.10%           |
> | Qwen3-30B-A3B          | 45.20%           | 68.30%              | 77.50%           |
>
> As shown in the table above, suppressing only a subset of experts can already lead to a substantial reduction in jailbreak behavior, with performance comparable to full expert fine-tuning. This result demonstrates the feasibility of a safety-enhancement strategy for MoE models that requires no additional retraining and relies solely on negative weight merging.
>
> **[2]** P. Chao et al., ​*JailbreakBench: An Open Robustness Benchmark for Jailbreaking Large Language Models*​, NeurIPS Datasets and Benchmarks Track, 2024.
>
> ---
>
> > **Q3:** Limited granularity in analysis: The focus on expert-level analysis, though aligned with MoE architecture, invites further scrutiny. If one were to analyze the safety attribution at the neuron level within experts, would the results be distinguishable from those observed in dense architectures? This could clarify whether the MoE-specific framing is essential or incidental.
>
> **A3:** We sincerely thank you for the valuable comment regarding the granularity of analysis. In this work, we chose to conduct safety attribution at the expert level because, in the MoE architecture, experts are explicitly activated functional units determined by the sparse routing mechanism. These experts directly influence the model's response behavior, making expert-level analysis more aligned with the structural properties of MoE compared to neuron-level analysis.
>
> Recent studies have also begun to investigate MoE architectures at the expert level, revealing potential security risks associated with the expert selection mechanism **[1] [2]**. We fully agree that the direction suggested by you is meaningful and worth further exploration. Notably, our proposed SAFEx framework is inherently extensible and can be adapted to support finer-grained interventions, such as neuron-level control. **According to your insightful comments, this has become a key direction of our ongoing research.** We will include additional clarification on this point in the revised manuscript. Once again, we sincerely appreciate your constructive feedback.
>
> **[1]** Hayes, J., Shumailov, I., & Yona, I. (2024). Buffer overflow in mixture of experts. ​*arXiv preprint arXiv:2402.05526*​.
>
> **[2]** Wang, Q., Pang, Q., Lin, X., Wang, S., & Wu, D. (2025). BadMoE: Backdooring Mixture-of-Experts LLMs via Optimizing Routing Triggers and Infecting Dormant Experts. ​*arXiv preprint arXiv:2504.18598*​.
>
> ---
>
> > **Q4:** Lack of actionable strategies for mitigation: While the work identifies structural weaknesses in MoE safety, it stops short of proposing concrete solutions or mitigations. Suggestions on how to improve expert routing, redundancy, or fallback mechanisms for safety-critical tasks would significantly strengthen the practical relevance of the study.
>
> **A4:** As discussed in the Q2, we have introduced a mitigation strategy targeting the MoE-related risks identified in our study and have demonstrated its effectiveness.

---

> > ### Author Response · Authors · 2025-08-02
> >
> > We truly appreciate your thoughtful comments, which have helped us clarify several important aspects of the work. We have carefully addressed your concerns in our point-by-point response, and we hope our explanations resolve the issues you raised. If you have any remaining questions or would be willing to share your thoughts on our clarifications, we would be sincerely grateful for your feedback.

---

> > ### Comment · Reviewer_cQJB · 2025-08-08
> >
> > Thank you for your response. My concerns have been addressed. Accordingly, I raise my rating to 4.

---

### Official Review · Reviewer_tAFC · 2025-07-01

**Clarity:** 3
**Significance:** 3
**Originality:** 3
**Rating:** 5
**Confidence:** 4

**Summary:**

This paper proposes the SAFEX framework to analyze positional vulnerability in MoE large language models. Using the SES algorithm, SAFEX identifies two types of safety-critical experts: those responsible for detecting harmful content and those controlling safe responses. Experiments show that mainstream MoE models heavily depend on a small set of experts for safety—disabling key experts significantly could reduce refusal rate.

**Questions:**

Please refer to the weaknesses outlined above.

**Ethical Concerns:**

["NO or VERY MINOR ethics concerns only"]

**Final Justification:**

The authors have fully addressed my concerns. Among them, two important points have been resolved, making me inclined to increase my score: (1) Figure 3 - the initial figure failed to illustrate the safety expert findings, but the updated explanation and quantitative results clarified the issue; (2) The safety control expert definition - their explanation was vivid and convincingly reasonable.

**Limitations:**

Yes

**Quality:**

3

**Strengths And Weaknesses:**

**Strengths:**

This study is the first systematic investigation into the security vulnerabilities of MoE architectures, providing a foundation for targeted safety alignment strategies. Observing differences in expert activation across various data types can help us better understand the model's internal mechanisms and training characteristics.

**Weaknesses:**

1) Regarding the drop in rejection rate after disabling experts: It seems unclear whether the performance degradation observed when disabling certain experts is due to their critical role in safety alignment (e.g., rejection capability) or if it reflects broader harm to the model's overall functionality.

2) Source of prompts in the Jailbreak Group: Could the authors clarify the origin of the adversarial prompts used in the Jailbreak Group? Were they generated from the MoE model itself (in a white-box setting) or from other models? This distinction is important for interpreting the observed expert activation patterns.

3) Expert activation differences across models: As noted in Line 195, "Jailbreak inputs significantly skew activations towards specific experts, highlighting potential safety vulnerabilities." However, in Figure 3, while there is a noticeable difference in deepseek-r1, I did not observe such clear distinctions in the other two models. The figure suggests that expert activation differences under jailbreak scenarios are relatively small for those models. Please provide further explanation or justification for these observations.

4) Definition of Safety Control Experts: Why are experts uniquely activated within the Regular group classified as Safety Control Experts? Intuitively, experts that effectively reject both regular and jailbreak prompts should be considered responsible for aligned safety generation. The current definition requires further clarification to align with this intuition.

---

> ### Author Rebuttal · Authors · 2025-07-30
>
> > **Q1:** Regarding the drop in rejection rate after disabling experts: It seems unclear whether the performance degradation observed when disabling certain experts is due to their critical role in safety alignment (e.g., rejection capability) or if it reflects broader harm to the model's overall functionality.
>
> **A1:** Thank you very much for your attention to the details of our experiments. In fact, one of the original purposes of constructing the $D_{Benign}$ dataset was to verify that the core capabilities of the model remain intact. Before conducting the mask experiments, we observed that all models used in the study were able to respond normally on $D_{Benign}$, with a refusal rate of 0%, as expected.
>
> In response to your review comments, we conducted a comprehensive evaluation of the masked models' outputs on $D_{Benign}$ using both manual inspection and model-based assessment. The results consistently showed that the selected models continued to produce valid responses, indicating that their basic question-answering functionality was preserved after masking. To further reduce potential confusion for future readers, we will include the corresponding experimental findings in the appendix.
>
> ​**In addition**​, to directly address your concern—“Will the masking downgrade the original performance of MoE?”—we conducted additional experiments on the AIME2024 dataset using **Qwen3-30B-A3B** and **Qwen1.5-MoE-A2.7B-Chat** (each prompted 5 times with the same question). The results are shown below:
>
> | Model                  | Pre-mask Score | Post-mask Score |
> | ------------------------ | ---------------- | ----------------- |
> | Qwen3-30B-A3B          | 77.4           | 75.2            |
> | Qwen1.5-MoE-A2.7B-Chat | 12.8           | 13.2            |
>
> The score fluctuations (–2.2 / +0.4) correspond to ±3 %, indicating no statistically significant change in mathematical-reasoning ability after masking. We therefore conclude that our masking strategy preserves overall performance while improving safety.
>
> ---
>
> > **Q2:** Source of prompts in the Jailbreak Group: Could the authors clarify the origin of the adversarial prompts used in the Jailbreak Group? Were they generated from the MoE model itself (in a white-box setting) or from other models? This distinction is important for interpreting the observed expert activation patterns.
>
> **A2:** We greatly appreciate your insightful question regarding the source of adversarial prompts in the Jailbreak Group. During the construction of this dataset, we applied a composite jailbreak strategy to rewrite $D_{Regular}$, aiming to enhance the prompts’ representativeness and difficulty. Specifically, we adopted ​**semantic paraphrasing**​, ​**adversarial perturbations**​, and **context reframing** to craft the adversarial versions.
>
> To clarify: all adversarial prompts in the Jailbreak Group were generated using Deepseek-R1, which is a non-target model. Therefore, the generation process falls under a black-box setting—we did not use the MoE model itself to generate or tune the prompts, nor did we assume access to routing paths or gradients. This design choice reflects a realistic threat model in which external adversaries do not have white-box access to proprietary models. Our intent was to simulate how a general-purpose attack, crafted independently of the MoE model’s internal structure, might still influence expert routing behavior.
>
> ---
>
> > **Q3:** Expert activation differences across models: As noted in Line 195, "Jailbreak inputs significantly skew activations towards specific experts, highlighting potential safety vulnerabilities." However, in Figure 3, while there is a noticeable difference in deepseek-r1, I did not observe such clear distinctions in the other two models. The figure suggests that expert activation differences under jailbreak scenarios are relatively small for those models. Please provide further explanation or justification for these observations.
>
> **A3:** We sincerely appreciate your insightful comments on our expert-activation analysis.
>
> * **Model clarification**
>
> All experiments were conducted on ​`deepseek-moe-16b-chat`, not ​*DeepSeek-R1*​. We have corrected this misreference.
>
> * **Why Figure 3 may look “flat”**
>
> Figure 3 only reports the *Top-N* experts in each layer. Within this head subset the bars appear relatively smooth; however, this perspective hides the broader skew present when **all 128 experts** are considered.
>
> * **Concrete numerical example drawn from Figure 3**
>
> Take **Qwen3-30B-A3B** on the Regular dataset: the tallest bar reaches **≈ 0.38** activation probability, whereas the theoretical mean is (k/K = 8/128 = 0.0625). Thus, the most active expert fires **~6 times** more often than chance. Under Jailbreak prompts the identities of the head experts shift, but their probabilities remain well above 0.30, still ∼5 times the mean, confirming a non-random redistribution rather than equalisation.
>
> * **Conservation argument**
>
> Because each token must activate exactly *k* experts, the aggregate activation count per layer is fixed. Therefore, whenever a few experts surpass the dashed theoretical mean, a larger number must fall below it. The apparently “flat” curves in the *Top-N* view simply omit these suppressed tail experts. This observation substantiates our claim that jailbreak inputs continue to bias routing toward specialised experts, even if that bias is distributed across a different subset.
>
> We have added the above concrete values and an explicit note beneath Figure 3 to pre-empt further confusion. Thank you again for helping us sharpen the presentation.
>
> ---
>
> > **Q4:** Definition of Safety Control Experts: Why are experts uniquely activated within the Regular group classified as Safety Control Experts? Intuitively, experts that effectively reject both regular and jailbreak prompts should be considered responsible for aligned safety generation. The current definition requires further clarification to align with this intuition.
>
> **A4:** We sincerely appreciate your scientifically grounded concern about our definition of ​**Safety Control Experts (SCEs)**.
>
> To address your question more intuitively, we first restate our academic motivation and then offer a concrete kitchen analogy that avoids circular reasoning.
>
> **1. Why we define SCEs by activation patterns, not by the final refusal signal**
>
> *Output-agnostic definition.* We deliberately characterize experts ​**solely by their activation patterns**​.
>
> * **Generalisability.** Activation tendencies remain stable across model architectures and fine-tuning recipes, whereas the surface refusal signal may change with system prompts or policy updates.
> * **Causal validity.** If we defined SCEs as “experts that appear in refused answers”, any later attempt to prove “masking SCEs ↓ refusal rate” would be circular, because “refusal” was already baked into the definition.
>
> **2. Kitchen analogy: the ‘safety MSG’**
>
> > Set-up. Imagine a chain restaurant with 64 jars of seasonings (experts).
> >
> > *Regular recipes* usually sprinkle a special **“safety MSG”** that keeps dishes within food-safety standards.
> >
> > *Jailbreak recipes* sometimes omit this MSG, so dishes may fail the inspection.
>
> Our procedure is therefore:
>
> 1. **Observation (output-free).** Without tasting the food (ignoring the refusal text), we record which jars are *always* used in regular dishes but *often* absent in jailbreak dishes.
> 2. **Intervention.** We hide this candidate jar and cook the same regular recipes. If the failure rate rises, the jar is causally linked to safety—exactly what we demonstrate in Figure 1 and Table 1.
>
> **3. Step-by-step comparison: why direct “refusal-based” tagging is circular**
>
> | Step | Direct refusal-based tagging                                      | Where the circularity lies                                                                                             |
> | ------ | ------------------------------------------------------------------- | ------------------------------------------------------------------------------------------------------------------------ |
> | ①   | Label every seasoning found in*passed*dishes as “safety MSG”. | “Safety” status**derives from**the pass result.                                                                |
> | ②   | Remove those seasonings and test if dishes now fail.              | Failure is expected—because the label was chosen*because*of passing—so this cannot prove​*seasoning → pass*​. |
>
> | Step | Our activation-based method                                                                     | Why it avoids the loop                                                   |
> | ------ | ------------------------------------------------------------------------------------------------- | -------------------------------------------------------------------------- |
> | ①   | Count usage frequencies; pick jars*always*present in regular but*rare*in jailbreak recipes. | Selection is independent of pass/fail.                                   |
> | ②   | Remove the jar and recook regular dishes.                                                       | If failure increases, we obtain genuine evidence for​*jar → pass*​. |
>
> We will integrate the above clarification into the final manuscript.

---

> > ### Comment · Reviewer_tAFC · 2025-08-02
> > **New review**
> >
> > The authors have addressed my concerns, particularly regarding the definition of safety control experts, and I decide to raise my score. Besides, I'm looking forward to an updated version of Figure 3 that can more clearly reflect the authors' findings.

---

> > > ### Author Response · Authors · 2025-08-02
> > >
> > > Thank you very much for your kind decision and thoughtful comments. We are truly grateful for your recognition and support. Due to the response policy, we are unfortunately unable to upload figures or PDF files at this stage. However, we will carefully revise and update Figure 3 in the next version to better convey our findings, as you suggested. We sincerely appreciate your valuable feedback and encouragement—it means a great deal to us.

---

### Official Review · Reviewer_gHUm · 2025-07-03

**Clarity:** 2
**Significance:** 3
**Originality:** 3
**Rating:** 4
**Confidence:** 4

**Summary:**

The paper introduces SAFEx, a novel analytical framework designed to investigate positional vulnerabilities in Mixture-of-Experts (MoE)-based large LLMs, where safety-aligned behaviors rely heavily on specific expert modules. SAFEx employs a three-stage pipeline: data collection to gather expert activation statistics, expert identification using a Stability-based Expert Selection (SES) algorithm to categorize experts into Harmful Content Detection Group (HCDG) and Harmful Response Control Group (HRCG), and expert validation via linear probing and masking experiments. Experiments on models like Qwen3-MoE demonstrate that disabling critical experts significantly reduces refusal rates for harmful inputs, confirming the positional vulnerability hypothesis.

**Questions:**

Please refer to the weaknesses, and if the authors could address my concerns, I would raise my rating.

**An extra suggestion:** While authors have stated that targeted defense mechanisms will be developed in the future, I believe incorporating potential solutions now would be better. (Not completing it will not drop my rating.)

**Ethical Concerns:**

["NO or VERY MINOR ethics concerns only"]

**Final Justification:**

My concerns have been addressed and thus, I raise my rating by 1.

**Limitations:**

yes

**Quality:**

2

**Strengths And Weaknesses:**

**Strengths:**
1. The paper pioneers the formalization of positional vulnerabilities in MoE-based LLMs, offering a novel perspective on safety alignment challenges.

2. The SAFEx framework effectively combines statistical analysis and validation to uncover expert-specific safety roles.

3. The structure of the paper is clear.


**Weaknesses:**

1. While I appreciate authors' exploration of positional vulnerabilities in MoE-based LLMs, the contributions still feel like not enough. Could the safety roles be further granulated to examine how different experts are sensitive to specific safety themes (e.g., hate speech vs. misinformation), and do different jailbreak techniques, such as adversarial suffixes versus semantic rephrasing, elicit distinct.

2. The division of experts into only two groups, HCDG and HRCG, seems overly prescriptive; it would be helpful to provide more justification for this binary categorization, perhaps by exploring whether additional functional groups or overlapping roles exist.

3. The experimental details lack clarity, particularly regarding the datasets used; specifying the names, sizes, and sources of these datasets, rather than relying on vague references to third-party data, would significantly enhance the reproducibility and credibility of the results.

4. Some typos, such as line 90, SES.

---

> ### Author Rebuttal · Authors · 2025-07-30
>
> > **Q1:** While I appreciate authors' exploration of positional vulnerabilities in MoE-based LLMs, the contributions still feel like not enough. Could the safety roles be further granulated to examine how different experts are sensitive to specific safety themes (e.g., hate speech vs. misinformation), and do different jailbreak techniques, such as adversarial suffixes versus semantic rephrasing, elicit distinct.
>
> **A1:** We greatly appreciate your insightful suggestion. Following your suggestions, we have designed and conducted additional experiments to further disentangle the safety roles of experts in MoE-based LLMs. Specifically, we curated a *diverse and balanced* dataset containing 500 prompts for each of five safety-relevant themes—hateful speech, gore, violence, pornographic and vulgar content—ensuring topic coverage within every category. Statistical analyses across these themes reveal the expert activation patterns, which are summarised in Table 1 below.
>
> | Model | Group | Hateful Speech | Gore | Violence | Pornographic | Vulgar Content | Intersection |
> | --- | --- | --- | --- | --- | --- | --- | --- |
> | Qwen3-30B-A3B | $\mathcal{E}_{\text{id}}$ | 26 | 20 | 34 | 18 | 21 | 9 |
> | Qwen3-30B-A3B | $\mathcal{E}_{\text{ctrl}}$ | 21 | 18 | 23 | 16 | 19 | 11 |
> | Qwen1.5-MoE-A2.7B-Chat | $\mathcal{E}_{\text{id}}$ | 24 | 31 | 23 | 19 | 27 | 10 |
> | Qwen1.5-MoE-A2.7B-Chat | $\mathcal{E}_{\text{ctrl}}$ | 17 | 15 | 20 | 34 | 11 | 7 |
>
> > “Intersection” denotes the number of experts jointly activated across all five safety themes within the same group.
>
> These results reveal that although the most frequently activated experts vary slightly across themes and jailbreak techniques, there remains a substantial overlap—especially within the Harmful-Content Detection Group (**HCDG**)—suggesting that pre- and post-training implicitly guide certain experts toward specialisation in unsafe content.
>
> We further analyzed model responses to three representative jailbreak strategies—semantic paraphrasing, adversarial perturbations, and context reframing—as summarised in Table 2 below.
>
> | Model | Group | Semantic Paraphrasing | Adversarial Perturbations | Context Reframing | Intersection |
> | --- | --- | --- | --- | --- | --- |
> | Qwen3-30B-A3B | $\mathcal{E}_{\text{id}}$ | 31 | 28 | 19 | 14 |
> | Qwen3-30B-A3B | $\mathcal{E}_{\text{ctrl}}$ | 41 | 29 | 23 | 14 |
> | Qwen1.5-MoE-A2.7B-Chat | $\mathcal{E}_{\text{id}}$ | 28 | 29 | 18 | 11 |
> | Qwen1.5-MoE-A2.7B-Chat | $\mathcal{E}_{\text{ctrl}}$ | 27 | 20 | 22 | 13 |
>
> Despite variations in the top-activated experts across jailbreak methods, a sizeable overlap persists between HCDG and the Harmful-Response Control Group (**HRCG**), suggesting shared internal pathways that adversaries may exploit.
>
> Many thanks for your insightful feedback. We will integrate the above analyses, tables, and corresponding discussions into the revised manuscript to strengthen our contributions.
>
> ---
>
> > **Q2:** The division of experts into only two groups, HCDG and HRCG, seems overly prescriptive; it would be helpful to provide more justification for this binary categorization, perhaps by exploring whether additional functional groups or overlapping roles exist.
>
> **A2:** We sincerely thank you for highlighting the potential limitations of our current binary categorization of safety-relevant experts. In the early stages of this study, we experimented with a *richer taxonomy* that distinguished, for example,
>
> **(i) harmless-content control experts,(ii) implicit-risk experts, and (iii) explicit-risk experts.**
>
> While these finer-grained groups appeared meaningful within a single architecture, they did not exhibit stable, transferable activation patterns across the four MoE models we investigated. By contrast, the two groups we ultimately report—Harmful Content Detection Group (HCDG) and Harmful Response Control Group (HRCG)—emerged consistently and displayed clear, complementary functional signatures in every model. For this reason, we focused our analysis and ablation studies on HCDG and HRCG, which best capture the cross-model regularities that are most actionable for the community. Looking ahead, we are developing an automated expert-group discovery framework that combines unsupervised clustering with LLM-based agents to reveal additional or overlapping functional groups under broader safety scenarios. We will append the above discussion into the paper to clarify our design choices and to outline this ongoing research direction.
>
> ---
>
> > **Q3:** The experimental details lack clarity, particularly regarding the datasets used; specifying the names, sizes, and sources of these datasets, rather than relying on vague references to third-party data, would significantly enhance the reproducibility and credibility of the results.
>
> **A3:** Thanks for your careful review. We will try our best to improve the paper’s readability. Specifically, we have summarized the distribution of harmful topics and the provenance of each dataset in Figure 1 of Appendix A. The datasets $D_{Regular}$, $D_{Jailbreak}$, and $D_{Benign}$ each comprise 500 examples. In Section 2.1 (Dataset Construction), we will add a pointer to Appendix A and explicitly state the size of each dataset.
>
> ---
>
> > **Q4:** Some typos, such as line 90, SES.
>
> **A4:** Thank you very much for your careful attention to the spelling details in our paper. We have thoroughly reviewed the manuscript and corrected the identified spelling errors.
>
> ---
>
> > **An extra suggestion:** While authors have stated that targeted defense mechanisms will be developed in the future, I believe incorporating potential solutions now would be better. (Not completing it will not drop my rating.)
>
> **A5:** We sincerely thank you for this constructive suggestion. Following your advice, we have conducted a set of preliminary experiments on potential defence strategies and report the most effective approach below.
>
> **Method.** Building on the safety-critical experts identified by SAFEX, we employ *negative-weight merging* to attenuate the toxic-generation capability of the HRCG experts. Concretely, we fine-tune either a subset or the entirety of these experts on the JBB-Behaviors dataset [1] to obtain toxicity-related weight deltas, and then subtract these deltas from the original Qwen-MoE checkpoints, yielding two safety-enhanced variants.
>
> **Results.** We evaluate the modified models on the $D_{Jailbreak}$ dataset and compare refusal rates before and after partial/full merges:
>
> | Model | Before Alignment | After Partial Merge | After Full Merge |
> | --- | --- | --- | --- |
> | **Qwen1.5-MoE-A2.7B-Chat** | 52.00 % | **74.80 %** | **80.10 %** |
> | **Qwen3-30B-A3B** | 45.20 % | **68.30 %** | **77.50 %** |
>
> Suppressing only a subset of HRCG experts already yields a marked improvement in robustness, approaching the effect of fully fine-tuning all experts. This demonstrates a lightweight, retraining-free safety-enhancement pathway for MoE architectures.
>
> **Integration into the manuscript.** We will briefly summarise these findings in **Section 4 (Conclusion)** and provide complete experimental details—including hyper-parameters, data splits, and ablation analyses—in the appendix.
>
> **[1]** P. Chao et al., ​*JailbreakBench: An Open Robustness Benchmark for Jailbreaking Large Language Models*​, NeurIPS Datasets and Benchmarks Track, 2024.

---

> > ### Comment · Reviewer_gHUm · 2025-08-05
> >
> > Thank you for your response. My concerns have been addressed and thus, I raise my rating by 1.

---

> > > ### Author Response · Authors · 2025-08-06
> > >
> > > We sincerely thank you for your thoughtful review and valuable feedback, as well as for raising your rating. We truly appreciate the opportunity for this exchange and are grateful for your recognition and support of our work.

---

> ### Author Response · Authors · 2025-08-02
>
> We sincerely appreciate your insightful comments and the time you dedicated to reviewing our manuscript. We have revised the paper accordingly and attempted to address all the points raised in your review. If any parts of our response remain unclear or if further clarification is needed, we would be truly grateful if you could let us know. Your additional input would be invaluable in helping us further improve the quality and clarity of the work.

---

### Note · Authors · 2025-08-12

We sincerely thank the Area Chairs, Program Chairs, and reviewers (gHUm, tAFC, cQJB, qRFt) for their time and efforts during the review process. The constructive feedback we received has significantly improved our work, and we are pleased that our detailed responses and new experiments led all reviewers to raise their scores, indicating that their concerns were addressed.

In this revision, we fully addressed reproducibility by adding comprehensive details on datasets, jailbreak methods, evaluation protocols, and hyperparameters, ensuring the work is transparent and verifiable. We conducted finer-grained experiments on five safety themes and three jailbreak techniques, which validated our core claims about expert specialization, and we clarified the activation-based, non-circular logic of our expert identification framework, strengthening its methodological soundness.

Importantly, we demonstrated that our safety interventions do not harm the overall performance of MoE models. Building on reviewer suggestions, we also moved beyond diagnosis to propose a practical mitigation strategy — a negative-weight merging technique — which proved highly effective.

The final manuscript offers a complete contribution: from identifying the novel "positional vulnerability" in MoEs to presenting a validated, low-cost, and actionable solution. We believe this work provides meaningful value to the community and closes the author–reviewer discussion on a strong and constructive note.

Best regards,


The Authors of Submission 28857

---

### Decision · Program_Chairs · 2025-09-17

**Decision:**

Accept (poster)

**Comment:**

This paper studies safety vulnerabilities in MoE-based LLMs and proposes SAFEx, a framework to identify safety-critical experts. Reviewers initially had significant concerns about missing experimental details, limited granularity in the analysis, and the lack of actionable mitigation strategies. The authors added clarifications, more fine-grained experiments, and a lightweight mitigation method, which reviewers agreed addressed their main points. While the framing and contributions remain somewhat incremental and the methodology not entirely novel compared to prior dense-model work, the paper is now technically solid and reproducible. I recommend acceptance.